# The Extended Cleavage Specificity of Channel Catfish Granzyme-like II, A Highly Specific Elastase, Expressed by Natural Killer-like Cells

**DOI:** 10.3390/ijms25010356

**Published:** 2023-12-26

**Authors:** Michael Thorpe, Srinivas Akula, Zhirong Fu, Lars Hellman

**Affiliations:** Department of Cell and Molecular Biology, Uppsala University, Box 596, SE-751 24 Uppsala, Sweden; thethorpefamily4@gmail.com (M.T.); srinivas.akula@icm.uu.se (S.A.); fuzhirong.zju@gmail.com (Z.F.)

**Keywords:** fish, serine protease, cleavage specificity, tryptase, NK cells, evolution

## Abstract

The extended cleavage specificity of catfish granzyme-like II has been characterized using substrate phage display. The preference for particular amino acids at and surrounding the cleavage site was further validated by using a panel of recombinant substrates. This serine protease, which has previously been isolated as cDNA from a catfish natural killer-like cell line showed a preference for Ala in the P1 position of the substrate, and for multiple basic amino acids N-terminally of the cleavage site. A closely related zebrafish serine protease (zebrafish esterase-like) showed a very similar cleavage specificity, indicating an evolutionary conservation of this protease specificity among various fish species. Two catfish serine proteases, originating from NK-like cells, have now been isolated and characterized. One of them is highly specific met-ase with similar characteristics as the mammalian granzyme M. This enzyme may be involved in the induction of apoptosis in virus-infected cells, with a potential target in (catfish) caspase 6. In contrast to catfish granzyme-like I, the second enzyme analyzed here does not seem to have a direct counterpart in mammalian NK cells, and its role in the immune function of catfish NK cells is, therefore, still not known. However, this enzyme seems to be able to cleave a number of cytoskeletal proteins, indicating a separate strategy to induce apoptosis in target cells. Both of these enzymes are very interesting targets for further studies of their roles in catfish immunity, as enzymes with similar specificities have also been identified in zebrafish.

## 1. Introduction

In mammals, serine proteases constitute major granule constituents of several hematopoietic cell types. In hematopoietic cells, these proteases, which all belong to the large family of chymotrypsin-related serine proteases, are stored in their active forms within cytoplasmic granules, ready for rapid release upon activation of the cell. They are expressed primarily by mast cells, basophils, neutrophils, cytotoxic T cells (CTLs), natural killer cells (NK cells), and in very small amounts in eosinophils, but not in B cells, macrophages, or dendritic cells. In mammals, serine proteases of this large gene family play important and diverse roles in a number of physiological processes, including blood coagulation, food digestion, fertilization, complement activation, immunity, and tissue repair [1]. They can also have very different primary and extended cleavage specificities related to their roles in immunity or other physiological processes. For example, mammalian mast cells express such proteases with chymotryptic and tryptic activity. Human and mouse neutrophils store enzymes with elastolytic, tryptic, and also an enzyme with chymotryptic activity [2,3,4,5]. Human cytotoxic T cells and NK cells express and store several granzymes with asp-ase, tryptase, chymase, and meth-ase specificities, as exemplified by granzymes B, A, K, H, and M, respectively [2]. A relatively detailed picture has also been established of how these enzymes have appeared and diversified during mammalian evolution [3,4,6,7,8,9,10,11,12,13,14,15,16]; however, the situation in reptiles, amphibians, and fish is still only very fragmentary [17,18,19,20,21,22,23,24].

All of the chymotrypsin/trypsin-related enzymes share a common mechanism for cleaving peptide bonds with three vital residues (chymotrypsinogen numbering): His57, Asp102, and Ser195 [25]. These key amino acids form a catalytic triad located within the substrate binding pocket (termed S1) and are typically made up of three other residues 189, 216, and 226 [25]. Together, these three residues of the S1 pocket form the specificity-conferring triplet that provides clues to the primary specificity of the serine proteases. These three residues have been very valuable in obtaining the first indication of the primary specificity of enzymes from mammals, reptiles, and even amphibians but have proved to give very little information for enzymes belonging to this family in fish, except for the fish homologs to mammalian granzymes A and K [3]. The sequence divergence seems to be too extensive to position the residues correctly during an alignment [2,3]. This has meant that the fish enzymes have to be analyzed by other means than just bioinformatic studies to obtain information concerning both their primary and extended specificities.

Most of the different fish serine proteases that are related to the mammalian hematopoietic serine proteases have been identified by bioinformatic screening of various fish genomes. However, a few have been identified through direct cDNA cloning from fish cells or cell lines. This latter procedure results in additional information concerning their cellular origin.

An initial screening of different cell lines from the channel catfish (*Ictalurus punctatus*) for hematopoietic serine proteases resulted in the identification of cDNAs for three proteases termed granzyme-like I, II, and III. These proteases are distantly related to numerous mammalian T-cell granzymes (B to H), mast cell chymases, and neutrophil proteases [17]. Two of the cDNAs were cloned from catfish NK-like cells and one from a macrophage-like cell. In fish, two NK-cell homologues exist: NK-like cells and non-specific cytotoxic cells (NCCs). The relationship between the two populations is not entirely clear, although one difference between the cell types is their localization, where NK-like cells are derived from peripheral blood and NCCs are tissue based. Granzyme-like I and II were cloned from mixed lymphocyte culture-derived NK-like cells, and the third member (granzyme-like III) from the autonomous macrophage cell line 42TA [17,26]. The exact role of these NK-like cells is not known, but they seem to act similarly to mammalian NK cells. Very little work has been carried out on them except for the nice study by Shen et al., 2002 [26].

The specificity of catfish granzyme-like I has previously been presented [18]. This enzyme was found to be a highly specific met-ase similar to mammalian granzyme M. By a screening with the consensus substrate profile, catfish caspase 6 appeared as a potential substrate. Cleavage of the identified region of caspase 6 showed that catfish granzyme I-like efficiently cleaved this region, indicating that caspase 6 may be an in vivo substrate for this enzyme [18]. Catfish granzyme-like I may thereby be directly involved in caspase-dependent apoptosis induction of virus-infected cells similar to human granzyme M [27].

A fourth granzyme in this species, termed granzyme-1 (CFGR-1), has been identified earlier and is expressed in NCCs [21]. Phylogenetically, this protease clusters with human and mouse granzyme A and K, as well as having Asp-Gly-Gly as the specificity conferring triplet (based on chymotrypsinogen numbering 189-216-226), suggesting tryptase activity [21]. Indeed, recombinant granzyme-1 cleaves a tryptase-specific synthetic peptide and likely contributes to cell cytotoxicity induction based on a chromium release assay [23].

The aim of this study is to characterize the catfish granzyme-like II based on its extended cleavage specificity. Chronologically, this is the third enzyme expressed in catfish NK-like cells, and this study is an attempt to understand, in more depth, the role NK-like cells have in fish immunity. We have also analyzed a closely related zebrafish protease, the SPA, to see if the specificities are similar. Similar specificities would indicate a conserved function in fish immunity. To further extend this analysis, we have also analyzed the specificity of a close homologue to catfish granzyme-like I from zebrafish, the zebrafish-AE-like.

## 2. Results

### 2.1. Phylogenetic Analyses of a Panel of Fish Proteases Related in Sequence to the Mammalian Hematopoietic Serine Proteases

Human and mouse hematopoietic serine protease sequences were used as query sequences to identify similar sequences in a large panel of vertebrate genomes in the NCBI database using the TBLASTN algorithm. The ensemble database was also later screened for related sequences from different fish species to obtain the best coverage of the various genomes included in this study. The amino acid sequences of the active proteases were aligned with several different programs to study the relatedness between the various proteases. As the alignments looked very similar, only the alignment using MAFFT and the MrBayse program was used, generating a likelihood phylogenetic tree that is depicted in Figure 1A. An enlarged version of the fish proteases clustering in a separate branch of the tree is shown in Figure 1B. The phylogenetic analysis was performed essentially as described in a previous publication using the same strategy and sequences [3].

Of these serine proteases, the most similar to catfish granzyme-like I was a zebrafish protease termed arginine esterase-like (ZF-AE-like), and to catfish granzyme-like II was the zebrafish SPA (Figure 1B). Arginine-esterase-like is just a name added by the bioinformaticians that set the layout of the genome as they run a homology search, and the gene they identify was the closest homologue gives the name in the annotation even if the protease functionally most often has no similarity. These names, therefore, often change when more information becomes available. As can be seen from Figure 1, the fish proteases form a separate subfamily in the large tree and mammalian homologues, thereby, giving little help in defining the function and name of the particular fish protease based on low similarity to a particular mammalian protease.

### 2.2. Production, Purification, and Activation of Catfish Granzyme-like II, Zebrafish Esterase-like (AE-like), and Zebrafish SPA

DNA constructs containing the coding regions for the active catfish granzyme-like II, zebrafish esterase-like (AE-like), and zebrafish SPA, all with an N-terminal His_6_-tag followed by an enterokinase (EK) site were designed and ordered from Genscript. These fragments were cloned into the mammalian expression vector pCEP-Pu2 for expression in HEK293-EBNA cells. The His_6_-tag facilitates purification on Ni^2+^ chelating immobilized metal ion affinity chromatography columns, and cleavage with EK activates the enzyme, whilst simultaneously removing the His_6_-tag and the EK site (Figure 2). Following purification, aliquots of the enzymes were activated by EK cleavage, resulting in a drop in molecular weight of 1–2 kDa (Figure 2).

### 2.3. Substrate Phage Display

To determine the extended cleavage specificity of catfish granzyme-like II, a phage T7-based system was used where individual peptide sequences are displayed on the surface of the phage. This system enables the characterization of a region covering both 4–5 amino acids upstream and downstream of the cleavage site. The library used had a complexity of approximately 50 million different peptide sequences. After seven selection rounds, the catfish granzyme-like II selected phages showed at least a three-order of magnitude increase compared to the PBS control. Phage display of the same enzyme was performed independently by three persons over a period of several years to validate the system, shown as rounds 1–3 in Figure 3. Phage plaques from the last selection round were picked, and the region encoding the peptide sequence was amplified using PCR, sequenced, and aligned. Each row represents an individually sequenced random region, and multiple similar sequences are shown to the right where necessary. The alignment showed a highly specific selection, with an apparent preference for several positively charged amino acids, primarily Arg in a row, two, three, or four, followed by three Ala (Figure 3B). There were also many substrates with two Arg separated by one other amino acid. This separation of amino acids did not seem to be important for substrate selectivity as there were a number of different amino acids in this position among the phage sequences (Figure 3). The three independent phage display runs showed very similar results, which clearly indicates the high reproducibility and robustness of the technique (Figure 3).

### 2.4. Peptide Cleavage Analysis

To determine the exact cleavage site within the selected sequences from the phage display analysis, three peptides were produced with different variants of the consensus regions from the phage display (Figure 3A,B). The peptides were cleaved with the catfish granzyme-like II enzyme and the cleavage products were analyzed by mass spectrometry (Figure 3B). After one hour, the full-length peptides were cleaved into two fragments, where the larger major peptide had the sizes of 699.458, 756.495, and 671.420 for the three original peptides, respectively (Figure 3B). The molecular weight of the resulting larger products reflected cleavage after an Ala in all three peptides, showing that the enzyme is an Ala-selective elastase with a strong preference for basic amino acids in positions N-terminally of the cleavage site (Figure 3B).

### 2.5. Phage Display Sequence Verification Using Recombinant Substrates

In order to validate the phage display sequence data and to address the importance of variations of amino acids in the aligned phages, a unique type of recombinant substrate has been developed in our lab. The recombinant substrates are based on two identical thioredoxin (trx) proteins separated by a short flexible kinker region of repeating Ser and Gly residues followed by the nine amino acid consensus region obtained from the phage display and variants. A number of such sequences were analyzed by the cleavage of recombinant substrates in this two-trx system (Figure 4A,B). The phage display analysis resulted in a number of sequences with two, three, or four Arg in a row or with two Arg residues separated by one amino acid, which appeared to have a lower importance for the specificity (Figure 3). A panel of different substrates based on the consensus cleavage site and different variants from these sequences was constructed to obtain quantitative information concerning the importance of a particular amino acid in and around the cleavage site. Almost no cleavage was observed, with a substrate having only one positively charged residue (SV**R**AAAG) (Figure 4C). A substrate with two adjacent Arg (VV**RR**AAAG) in a row showed efficient cleavage, and substrates with three (VV**RRR**AAAG) or four Arg was cleaved even better (Figure 4C,D). Substrates with two Arg residues separated by one amino acid (VV**R**V**R**AAAG) were also efficiently cleaved by this enzyme (Figure 4D). However, when changing the P1 residue from an Ala to a Leu no cleavage occurred, showing the high specificity for Ala in the P1 position (Figure 4D). A Pro or a Gly seemed to be tolerated in the P1 position (Figure 5), or more likely, the P1 position shifts to being in the P2 position to move an Ala into the P1 position, as can be seen from peptide cleavage of the substrate with only two Arg residues (Figure 3B). However, a Phe in this position did not seem to be tolerated, or when two Val following the first Ala were included (Figure 5). The separating amino acid, when having substrates with two Arg separated, appeared to tolerate most amino acids except Pro, as seen in Figure 3A and Figure 5.

### 2.6. Analysis of the Cleavage Specificity of Zebrafish SPA

Zebrafish SPA is the most closely related protease in zebrafish to catfish granzyme-like II. Therefore, a comparison of the cleavage specificity was made. By using the same set of recombinant substrates as for the analysis of catfish granzyme-like II, a similar cleavage specificity was seen (Figure 6), although the zebrafish SPA was slightly less specific. Zebrafish SPA tolerated, to a minor extent, a Pro in between the two Arg residues in the Arg-X-Arg substrates and also a Leu following the three Arg residues in the Arg-Arg-Arg-Leu substrate (Figure 6).

### 2.7. Analysis of the Cleavage of Caspase 6 by Zebrafish AE-like

In a previous study, the catfish granzyme-like I did not cleave zebrafish caspase 6 [18]. To see if the cleavage was conserved over the species barrier for this protease, the cleavage activity of the corresponding zebrafish enzyme on zebrafish caspase 6 was analyzed [18]. Based on the phylogenetic tree (Figure 1), the most closely related zebrafish enzyme to catfish granzyme-like I was zebrafish AE-like. Therefore, this enzyme and the two 2xTrx substrates having the potential cleavable region for zebrafish and catfish caspase 6 were produced (Figure 7). As seen in Figure 7, the figure zebrafish AE-like cleaved both of these sequences, although the zebrafish sequence had lower efficiency.

### 2.8. Screening for Potential In Vivo Substrates

Screening for potential in vivo targets was confined to protein sequences from the channel catfish with a few variants of the consensus sequences obtained from the phage display analysis (RRRAAA, RRRA, RRAA, RVRA, and RRGA) (Figure 3). A number of potential targets were identified, the majority being connected to intracellular cytoskeleton proteins. A selection of these potential targets is presented in Table 1. Among the targets, both actin–myosin and tubulin interacting proteins were identified, indicating the breakdown of the intracellular cytoskeleton may be an additional target to enhance apoptosis of target cells by catfish granzyme-like II, together with activation of caspases by catfish granzyme-like I (Table 1).

### 2.9. Screening for Similar Protease Specificity

Due to very limited previous knowledge about the catfish granzyme-like II, plus its distinct extended specificity shown here, the MEROPS database was screened for other proteases with similar specificities. When fixing at least four positions (from P3-P2′), no recorded protease with similar specificity was found.

### 2.10. The Genomic Loci Encoding the Catfish Granzyme-like I, II, and III

When the studies on the evolution of fish hematopoietic serine proteases were initiated by analyzing the cleavage specificity for catfish granzyme-like I, no information was available on the catfish genome. However, during the intermittent years, more information has become available. Therefore, looking closely at the loci encoding the three catfish enzymes initially isolated from in vitro cultured catfish NK-like and macrophage-like cells is now possible. The catfish granzyme-like enzymes I, II, and III are located in two different genomic locations: catfish granzyme-like I and catfish granzyme-like II in one region, the fish met-ase locus and catfish granzyme-like III in a separate locus with no resemblance to any of the mammalian loci encoding hematopoietic serine proteases (Figure 8).

## 3. Discussion

A relatively detailed picture of the evolution of the hematopoietic serine proteases has been established for tetrapods and for one of the loci, the granzyme A/K locus, for all vertebrates. The first enzymes to appear during vertebrate evolution are the enzymes of the granzyme A/K subfamily, which are found in essentially all studied vertebrate species, from cartilaginous fish to humans. The majority of these granzyme A/K enzymes also appear to have relatively similar tryptic cleavage specificities. However, there are exceptions. We have recently characterized a new member of this family from a cichlid, the *Zebra mbuna*, where one of the granzyme A/K homologs has changed primary specificity and is now a highly specific chymase with selectivity for Tyr in the P1 position [28].

Of the other serine proteases, the apoptosis-inducing granzyme B of cytotoxic T cells and NK cells seems to be one the first enzymes to appear out of the classical hematopoietic serine proteases. This is based on its identification in a frog, with an active site that appears from in-silico analysis to have asp-ase or glu-ase activity [6]. A granzyme B homologue is also clearly present in reptiles, as seen in the Chinese alligator, as well as in monotremes, as exemplified by the platypus and also in a marsupial, the American opossum [14,15,20]. In these three species, there is also evidence for a classical mast cell chymase and/or a cathepsin G homologue [14,15,20]. An early ancestor of the neutrophil proteases, proteinase 3/N-elastase, also seems to appear in amphibians [4]. However, based on sequence analysis, it is not possible to determine whether it is more proteinase 3-like or N-elastase-like, but after initial cleavage specificity analysis, it appears to be more closely related to proteinase 3 [4]. The mast cell chymase does not seem to be present in frogs, indicating that it first appeared with the reptiles [6].

Compared to the tetrapods, the view of the corresponding expansion of these proteases in various fish species is still only fragmentary. Relatively few fish proteases have been characterized, and the primary and extended specificities differ substantially from the patterns we see in tetrapods. The enzymes that seem to be best conserved are ones closely related to mammalian granzymes A and K [3]. The majority of them are, as the mammalian enzymes encoded from the granzyme A/K locus, where they appear to have trypsin-like primary specificity, with a preference for Arg over Lys in the P1 position of substrates [3]. They also seem to be expressed by similar cells, including cytotoxic T cells and NK cells. The catfish granzyme-like II appears to differ from some other fish proteases by having at least one granzyme A/K homologue with proven tryptase activity, which is present in a new position in the genome. This position is not related to any of the previously identified loci for hematopoietic serine proteases and additionally seems to be the only serine protease gene in that region of the genome (Figure 8C).

Concerning the catfish granzyme-like I and II, both are located in the met-ase locus and this locus has a similar organization as the corresponding human locus (Figure 8A). However, the enzyme specificities markedly differ. Although catfish granzyme-like I has a primary specificity for Met, which matches the human granzyme M, which is located in the same locus, the extended specificity is very different and also appears to be much more specific in fish compared to the human enzyme. However, both seem to be expressed by NK- or NK-like cells. This pattern is also similar when studying the primary specificities for catfish granzyme-like II. This enzyme is an elastase, similar to both human N-elastase and proteinase 3, but the cell origin and extended specificities differ, where catfish granzyme-like II is expressed by fish NK-like cells, in comparison to both N-elastase and proteinase 3, which are expressed by human neutrophils. The extended specificity is also much more specific in the fish enzyme than the two human enzymes, indicating a more limited number of potential in vivo substrates for the fish enzyme.

The primary in vivo targets for the different fish proteases are also not known. However, for catfish granzyme-like I, caspase 6 is a potential candidate [18]. This enzyme can cleave a region in caspase 6 that corresponds in position to the site in human caspase 3, which is the primary activation site for human granzyme B [18]. However, the role of the catfish enzyme in potential caspase 6 activation has not been proven due to difficulties in producing catfish caspase 6 as a recombinant protein. The in vivo targets for catfish granzyme-like II are also not known. However, the screening with several variants of the consensus cleavage site obtained by phage display of the entire catfish proteome has resulted in an array of different cytoskeletal proteins, indicating that degradation of the cytoskeleton may be an alternative or additional mechanism of inducing apoptosis of target cells (Table 1). Additional work on these potential mechanisms for apoptosis induction of target cells by catfish NK-like cells is needed to clarify the questions that still remain concerning these enzymes.

By looking at similar proteases in catfish and zebrafish, it is possible to see that the specificities of both catfish granzyme-like I and II have been relatively well conserved over what has been estimated to be 110–160 million years; the estimated time of divergence between these two fish species [29]. The similar cleavage specificities of catfish granzyme-like I and zebrafish AE-like, as well as catfish granzyme-like II and zebrafish SPA, thereby indicating that these enzymes cleave important targets contributing to the roles of NK-like cells in fish.

Several of the mammalian enzymes have a relatively broad specificity, including the majority of neutrophil proteases as well as the mast cell chymase, whereas the fish proteases, including the catfish granzyme-like I and II show a very high selectivity of amino acid sequence. Therefore, catfish granzyme-like I and II most likely cleave a relatively limited number of substrates compared to the mammalian enzymes. It will be interesting to see if this view is also valid when more fish proteases are added to the list of fully characterized enzymes. An analysis of a larger number of different hematopoietic serine proteases from fish will hopefully increase our understanding of the evolutionary processes that have participated in the generation of a complex set of hematopoietic serine proteases as an important part of vertebrate immunity, the conserved targets of central importance for vertebrate immunity as well as the role of these protease subfamilies in fish immunity. By identifying conserved targets between these proteases in fish and mammals, we can also find out the most essential functions of these proteases in vertebrate immunity.

## 4. Materials and Methods

### 4.1. Phylogenetic Analyses

The phylogenetic analysis was performed as described in a previous publication using the same strategy and sequences [3]. Sequences relating to catfish granzyme-like II were systematically uncovered by BLASTp searching of all animal NCBI databases. The mature catfish granzyme-like II was used as the query sequence, and all novel derived sequences were analyzed using the multiple alignment program MAFFT with G-INS-i strategy and default parameters to determine whether they belonged to the serine protease family. To visualize the relationship between catfish granzyme-like II and those from other species, a phylogenetic tree using the Bayesian interference of phylogeny algorithm with posterior probabilities in the MRBAYES program was constructed and viewed in FigTree (v1.4). The amino acid sequences for mature proteins of serine proteases branching with catfish granzyme-like II were aligned using MAFFT.

### 4.2. Production of Recombinant Catfish Granzyme-like II, Zebrafish AE-like, and Zebrafish SPA

The channel catfish granzyme-like II sequence (GenBank accession numbers: (AY942182) (XP_017334488)) and two zebrafish serine proteases, one closely related to catfish I, the zebrafish AE-like (GenBank accession number: (XP_687163)) and one that is closely related to catfish II, the zebrafish SPA (GenBank accession number: (XP_003201101)) were designed and ordered from GenScript (Piscataway, NJ, USA). The synthesized constructs were cloned in the pU57 cloning vector, containing EcoRI and XhoI sites, and subsequently transferred to a pCEP-Pu2 vector, used for expression in mammalian cells [30]. The enzymes were produced as an inactive recombinant protein, with an N-terminal His_6_-tag followed by an enterokinase (EK) site. HEK 293-EBNA cells were grown to 70% confluency in a 25 cm^3^ tissue culture flask (BD VWR) with Dulbecco’s Modified Eagles Medium (DMEM) (GlutaMAX, Invitrogen) supplemented with 5% fetal bovine serum (FBS) and 50 µg/mL gentamicin. Following DNA (25 µg of granzyme-like II in pCEP-Pu2) transfection with lipofectamine (Invitrogen, Carlsbad, CA, USA), puromycin was added to the DMEM (0.5 µg/mL) to select for cells which had taken up the DNA along with heparin (5 µg/mL). Cells were expanded, and conditioned media was collected.

To purify the recombinant enzymes, 750 mL conditioned media was filtered (Munktell 00H 150 mm, Falun, Sweden) and 500 µL nickel nitrilotriacetic acid (Ni-NTA) agarose beads were added (Qiagen, Hilden Germany). The media with Ni-NTA beads were rotated for 45 min at 4 °C. Subsequently, the Ni-NTA beads were collected by centrifugation and transferred to a column containing a glass filter (Sartorius, Goettingen, Germany). After washing with PBS tween 0.05% + 10 mM imidazole + 1 M NaCl, the recombinant protein was eluted in PBS tween 0.05% + 100 mM imidazole fractions. The first fraction volume was half the Ni-NTA bead width (200 µL), and further fractions were eluted with a full bead width (400 µL). Individual fractions were run on SDS-PAGE gel, their concentrations estimated from a bovine serum albumin standard (BSA), and the most concentrated were pooled and kept at 4 °C.

### 4.3. Activation of Recombinant Catfish Granzyme-like II, Zebrafish SPA, and Zebrafish AE-like

The recombinant catfish and zebrafish enzyme concentrations were determined by SDS-PAGE, and the level of enterokinase (EK) (Roche, Mannheim, Germany) was adjusted for activation of the enzyme. A relative concentration was activated depending on when it was needed, where, for example, 70 µL of the eluted recombinant enzyme was digested with 1 µL EK for 3 h at 37 °C. The activated fractions were stored at 4 °C until use.

### 4.4. Substrate Phage Display

A T7 phage library containing 5 × 10^7^ variants, each displaying a unique nine amino acid sequence was used to determine the extended cleavage specificity of the catfish granzyme-like II enzyme. The nine amino acid region has been inserted into the C-terminal of the capsid 10 protein, followed by a His_6_-tag. Approximately 10^9^ plaque-forming units (pfu) were bound to 125 µL Ni-NTA agarose beads via their His_6_-tags for 1 hr at 4 °C with gentle rotation. Unbound phages were removed by washing ten times with PBS tween 0.05% + 1 M NaCl, followed by two washes with PBS. The beads were resuspended in 375 µL PBS and approximately 250 ng of recombinant catfish granzyme-like I was added. This reaction was incubated overnight or approximately 16 h at 37 °C with gentle rotation, allowing cleavage of the susceptible phages and their subsequent detachment from the Ni-NTA beads. From this, the supernatant containing released phages was recovered after centrifugation. Thirty µL was used in a plaque assay to determine the number of released phages. Briefly, ten-fold serial dilutions were made, mixed with (*E. coli*) BLT5615 (for propagation and visualization of plaques on a bacterial lawn) and plated on LA-Amp (50 µg/mL) plates, incubated for 2.5 h at 37 °C and then counted. The remaining supernatant was added to 10 mL BLT5615 bacteria (OD_600_ 0.5) for 75 min at 37 °C for phage expansion. From this, 1.5 mL was centrifuged to remove bacterial debris and 800 µL was transferred to a microcentrifuge tube containing 100 µL PBS and 100 µL 5 M NaCl. A sample of 125 µL Ni-NTA beads was added to the solution and placed at 4 °C for 2 h under rotation to allow phages to bind and thereafter, after washing, to start of the next selection cycle. The complete process was repeated a further 6 times, constituting 7 selection rounds. Individual plaques were isolated from the final selection round in 100 µL phage buffer before vortexing for 30 min and stored at 4 °C. The random nine amino acid regions contained in these phages were amplified by PCR (T7Select primers, Novagen, Sacramento, CA, USA) and sequenced by Eurofins (Ebersberg, Germany). The resulting sequences were translated using CLC viewer and aligned using Adobe Illustrator. A parallel control reaction without enzyme (only PBS) was also run under the same conditions and plaque numbers were compared to the enzyme sample.

### 4.5. Phage Display Sequence Verification Using a Two-Thioredoxin Approach

In order to verify the phage display data, a recombinant two-thioredoxin (2xTrx) system developed in our laboratory was used. Here, the random nine amino acid cleaved region determined from the phage display was introduced between two adjacent trx (from *E. coli*) proteins. Originally, a pET21 vector containing a single trx protein was modified to contain a second trx with BamHI and SalI sites in the intervening region. Here the random region was synthesized as oligonucleotides (Sigma, St Louis, MI, USA), which were ligated and inserted between BamHI and SalI restriction sites. This resulted in a vector containing a first trx followed by the random cleavable region and then a second trx with His_6_-tag (facilitating purification).

This construct was expressed in *E. coli* Rosetta gami (Novagen). Ten mL of an overnight culture was added to 90 mL LB+Amp (50 µg/mL) and 500 µL 20% glucose. After approximately 1 h (reaching OD_600_ 0.5), 100 mM isopropyl β-D-1-thiogalactopyranoside (IPTG) was added and the culture was placed on a shaker (moderate shaking) at 37 °C for 3 h. The culture was pelleted by centrifugation at 10,000 rpm for 3 min and the supernatant was discarded. The pellet was washed in 10 mL PBS tween 0.05%, centrifuged and pelleted again, followed by resuspension in 1/100th starting volume (i.e., 1 mL) PBS. To obtain the intracellularly expressed protein, the resuspended pellet was sonicated for 6 × 30 s on ice. The supernatant was transferred to a new microcentrifuge tube after centrifugation at 10,000 rpm for 10 min at 4 °C. To purify, 125 µL Ni-NTA beads were added and incubated for 45 min at 4 °C with gentle incubation. The solution with Ni-NTA beads was transferred to a 2 mL column (Terumo, Leuven, Belgium) containing a glass filter (Sartorius, Goettingen, Germany) and subsequently washed with 3 × 2 mL and 2 × 1 mL PBS tween 0.05% + 10 mM imidazole. To elute, fractions were collected after passing through PBS tween 0.05% + 100 mM imidazole. The first fraction volume was half the Ni-NTA bead width (75 µL), and further fractions were eluted with a full bead width (150 µL). Individual fractions were run on SDS-PAGE gel, their concentrations estimated from a BSA standard (and Bradford assay (Bio-rad, Hercules, CA, USA)), and the most concentrated were pooled and kept at 4 °C.

For cleavage analysis, approximately 250 ng of recombinant catfish granzyme-like II was added to 20 µg of the pooled 2xTrx protein (containing different sequences based on the phage display data), and aliquots of 5 µg removed after 0, 15, 45, and 150 min after enzyme addition. The reactions were run at room temperature, and the time point aliquots were analyzed on SDS-PAGE gel under denaturing conditions using pre-cast 4–12% Bis-Tris gels (Invitrogen, Carlsbad, CA, USA) and 1x MES buffer (Invitrogen, Carlsbad, CA, USA). Gels were stained with Coomassie colloidal solution to visualize the protein bands [31].

## Figures and Tables

**Figure 1 ijms-25-00356-f001:**
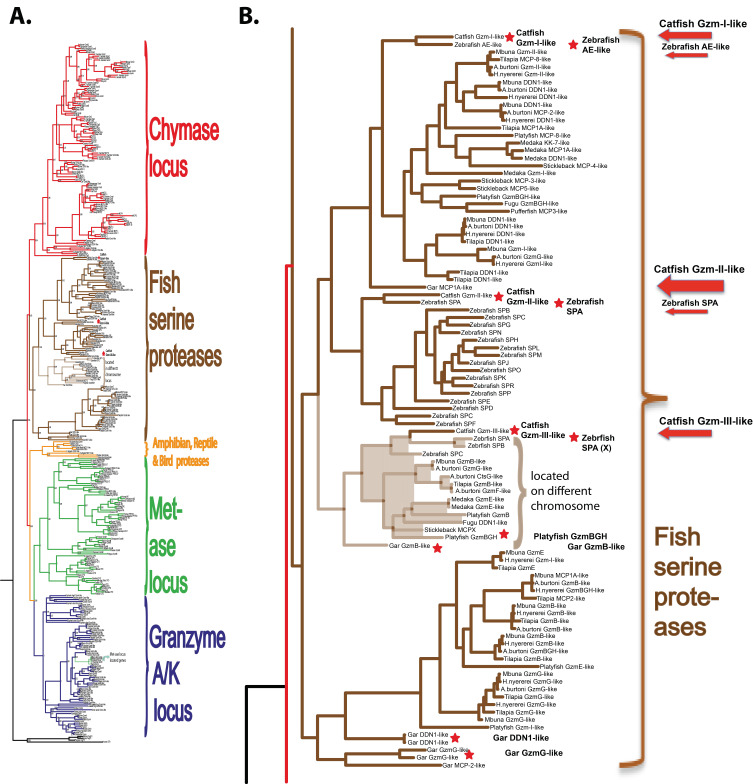
Phylogenetic relationship between catfish granzyme-like II and other hematopoietic serine proteases. All sequences were run in the multiple alignment program MAFFT to verify they belonged to the serine protease family. The tree was constructed using MRBAYES with a Bayesian interference of phylogeny algorithm (with posterior probabilities), opened with FigTree (v1.4), and annotated in Adobe Illustrator (CS5). Panel (**A**) shows the entire analysis involving a total of 368 vertebrate serine protease sequences. Panel (**B**) shows an enlargement of the branch of the major tree where the majority of the fish proteases are found, except the granzyme A/K-related fish proteases. The proteases of particular interest for this study are marked with red arrows. All enzymes for which we have produced recombinant proteins are marked with red stars.

**Figure 2 ijms-25-00356-f002:**
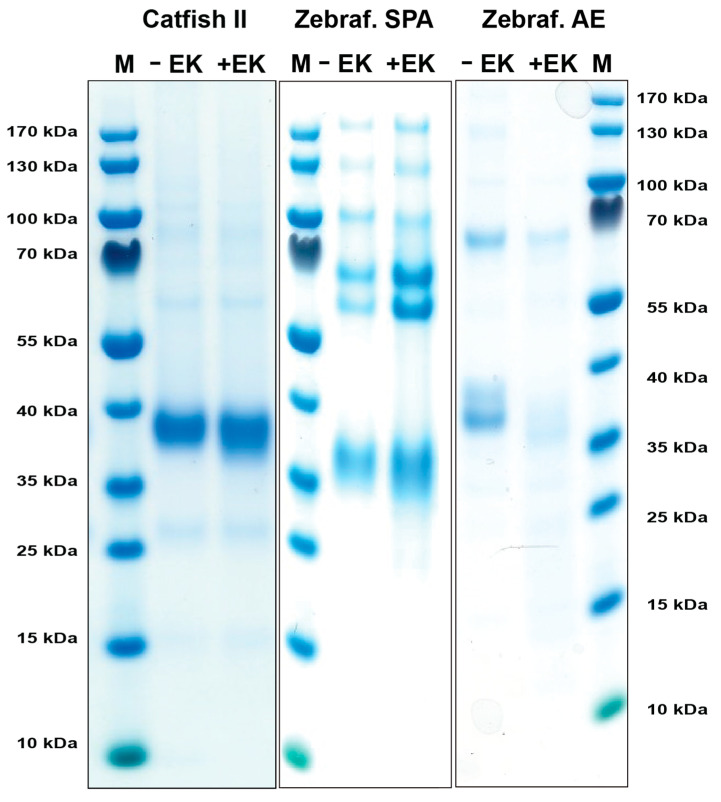
Recombinant catfish granzyme-like II, zebrafish SPA, and zebrafish AE-like. The enzymes were produced as an inactive protein (**left** lane) in HEK293-EBNA cells with a N-terminal His6-tag and enterokinase (EK) site, facilitating purification and activation, respectively. The addition of EK cleaves the N-terminal sites, resulting in an active enzyme and a subsequent drop in size (**right** lane). The enzymes were run on a 4–12% pre-cast SDS-PAGE gel and stained with Coomassie brilliant blue.

**Figure 3 ijms-25-00356-f003:**
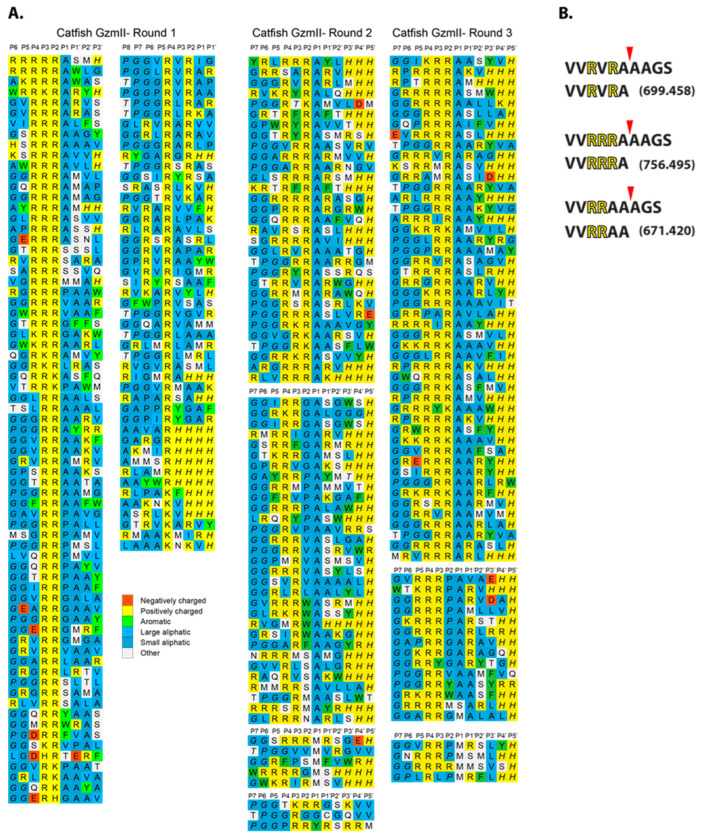
Substrate phage display sequences. A T7 phage library was subject to seven selection rounds with the catfish granzyme-like II protease. After the final round, phage plaques were isolated, and their random region was amplified and sequenced by PCR. The sequences were decoded and aligned. This procedure was performed independently by three different persons over a period of years between the runs. All three individual runs are presented separately in panel (**A**). Synthetic peptides were used to determine the exact cleavage site. The peptides were analyzed on SDS-PAGE gels after 1 h of digestion with the catfish granzyme-like II enzyme. As seen from panel (**B**), the cleavage occurred after an Ala residue, one or two positions C-terminally of the basic amino acids.

**Figure 4 ijms-25-00356-f004:**
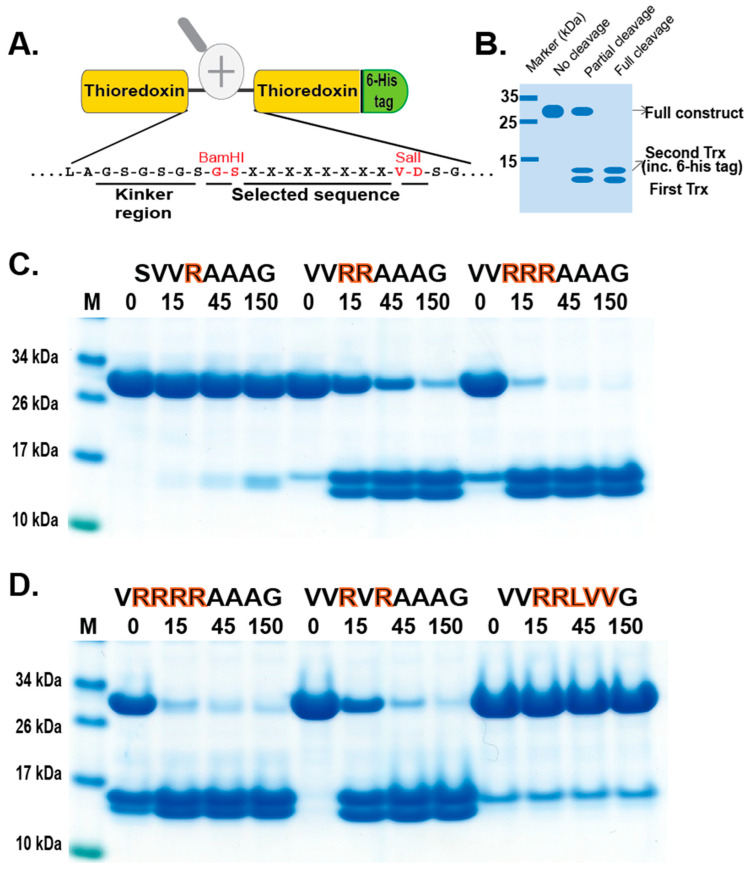
Verification of phage display sequences using the 2xTrx system. A number of phage display-derived sequences and variants of these sequences were added in between two adjacent trx proteins (panel (**A**)), expressed in *E. coli,* and subjected to catfish granzyme-like II (panels (**C**,**D**)). The results were run on pre-cast 4–12% SDS-PAGE gels. Hypothetical cleavage is shown (panel (**B**)) to highlight possible cleavage patterns. The individual lanes represent various time points after the addition of the enzyme in minutes. In this system, the samples act as internal positive and negative controls. When we take a sequence selected as preferred from the phage display, the cleavage is very efficient, and when we modify the sequence to separate from the consensus, we often see no cleavage. These two examples act as positive and negative controls, as can be seen from Figure 4C. Having only one Arg and no cleavage and having four Arg, we see almost complete cleavage already after 15 min.

**Figure 5 ijms-25-00356-f005:**
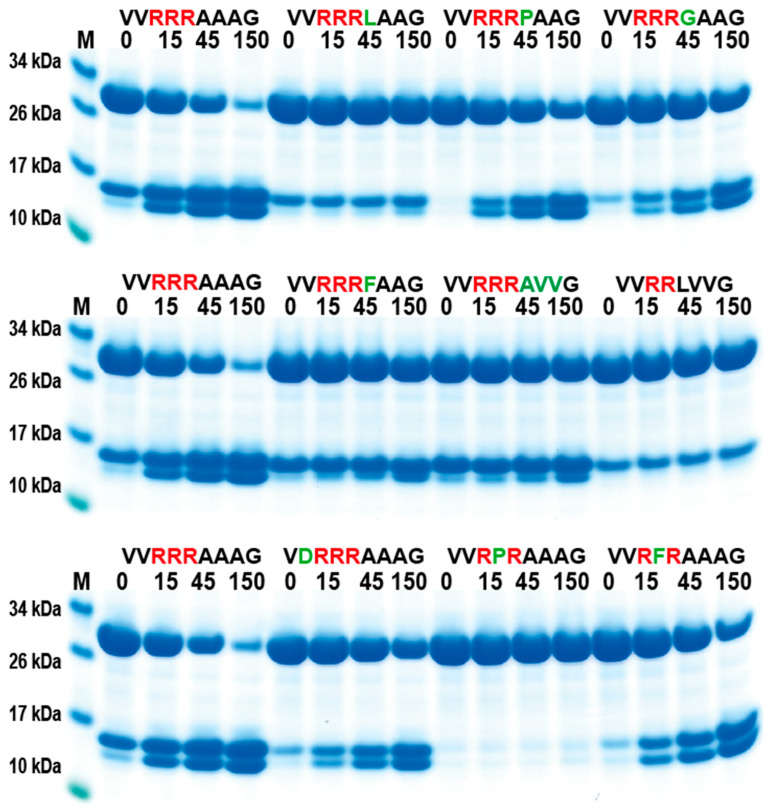
Verification of phage display sequences using the 2xTrx system. A number of phage display-derived sequences and variants of these sequences were added in between two adjacent trx proteins, expressed in *E. coli* and subjected to catfish granzyme-like II. The results were run on pre-cast 4–12% SDS-PAGE gels. The individual lanes represent various time points after the addition of the enzyme in minutes.

**Figure 6 ijms-25-00356-f006:**
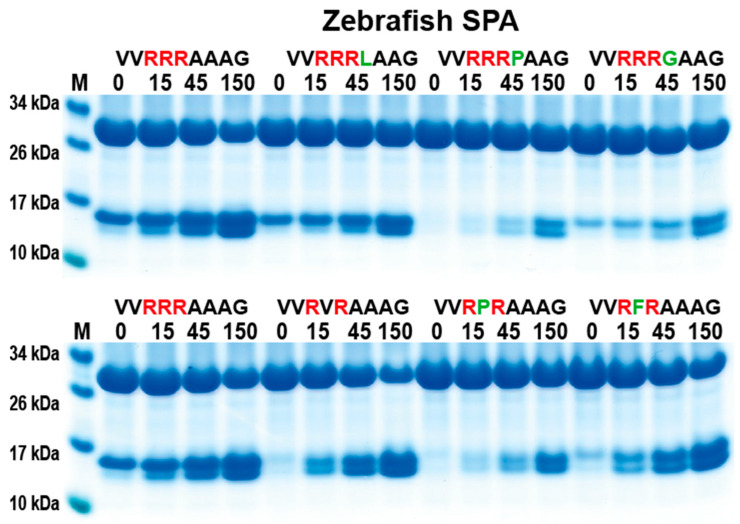
Analysis of the cleavage specificity of zebrafish SPA using the 2xTrx system. A number of the 2xTrx substrates used for the analysis of catfish granzyme like II were used to analyze the specificity of the closely related zebrafish enzyme, the zebrafish SPA (Figure 1).

**Figure 7 ijms-25-00356-f007:**
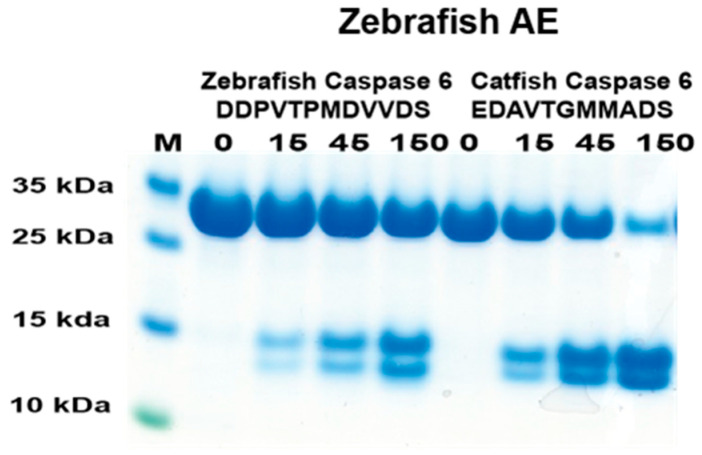
Analysis of the cleavage specificity of zebrafish AE-like using the 2xTrx system. A number of the 2xTrx substrates used for the analysis of catfish granzyme-like I were used to analyze the specificity of the closely related zebrafish enzyme, the zebrafish AE-like (Figure 1).

**Figure 8 ijms-25-00356-f008:**
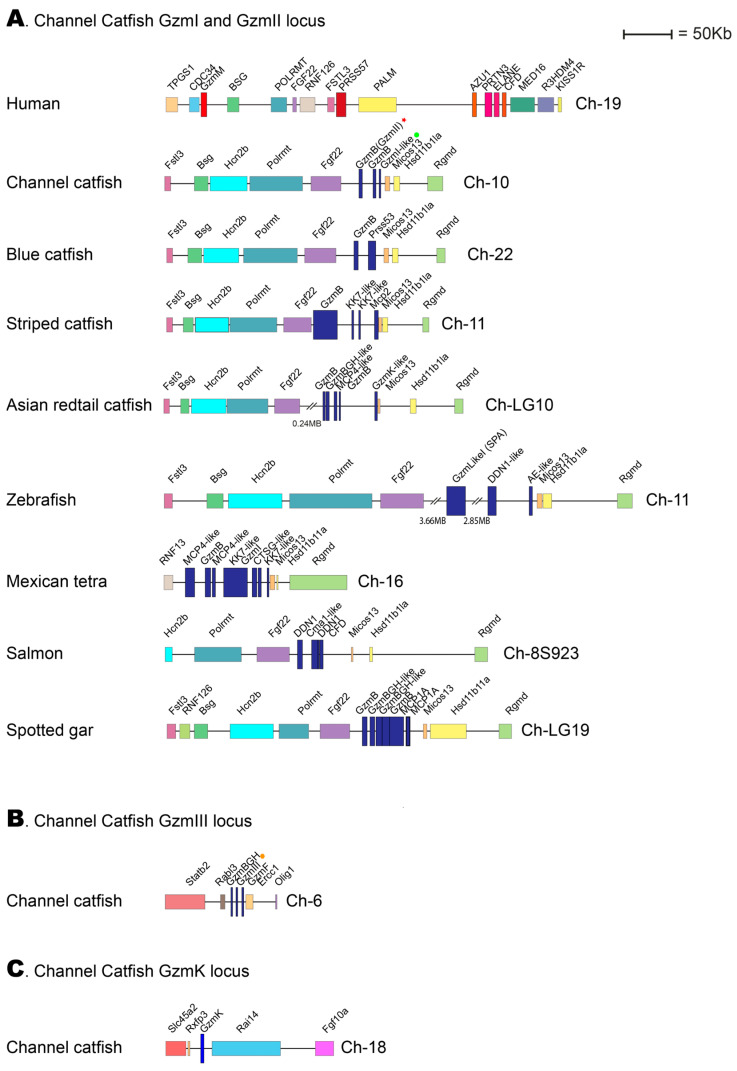
Genomic loci encoding catfish granzyme-like I, II, III, and catfish granzyme K. The loci encoding four different hematopoietic serine proteases from the channel catfish are presented in panels (**A**–**C**). The locus of particular interest for this study, the locus encoding catfish granzyme-like I and II have been analyzed in a panel of different fish species, as shown in panel (**A**). Only the catfish locus for granzyme-like III and for catfish granzyme K are presented in panels (**B**,**C**), respectively. The loci are shown in scale, and the sizes can be obtained by comparing them with the size bar. The catfish enzymes granzyme-like I, II, and III are marked with a green dot, a red star, and an orange dot, respectively, in the figure.

**Table 1 ijms-25-00356-t001:** Potential targets for catfish granzyme-like II. The major intracellular potential targets identified from a screening of the catfish proteome using the following short amino acid sequences originating from the consensus target sequence of catfish granzyme-like II, RRRAAA, RRRA, RRAA, RVRA, and RRGA.

**RRRAAA**
Shogushin isoforms (guides chromosome cohesion during cell division bundling microtubules)
Supervillin isoforms (possibly act as high-affinity link between actin filaments and plasma membrane)
**RRRA**
Dystonin isoform X18 (microtubule binding, cytoskeleton organization, and intracellular transport)
Nebulin isoforms (A large actin-binding protein)
**RRAA**
Microtubule-actin cross-linking factor 1 isoforms
Plectin isoforms (link between actin, microtubule, and intermediate filaments)
Unconventional myosin IXAb isoforms (perform key roles in a broad range of fundamental cellular processes)
**RVRA**
Nesprin 2 isoforms (Nuclear outer membrane component that binds actin filaments)
Cytoplasmic dynein 2 isoforms (drives the movement of cargoes along microtubules within cilia)
Microtubule-associated protein 1A isoforms (involved in microtubule assembly)
Filamin C isoforms (crosslink actin filaments into orthogonal networks)
Filamin B isoforms (cross-linking of actin to allow direct communication between the cell membrane and cytoskeletal network)
Myosin 10 isoforms (an actin-based motor protein)
**RRGA**
Microtubule-actin cross-linking factor 1 isoforms
Obscurin isoforms (may have a role in the organization of myofibrils)
Heat repeat-containing protein 5A isoforms (predicted to be involved in endocytosis)

## Data Availability

All data of importance for this study is available in this publication.

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
