# Peer review of "The Extended Cleavage Specificity of Channel Catfish Granzyme-like II, A Highly Specific Elastase, Expressed by Natural Killer-like Cells"

_ijms, 2023, doi:10.3390/ijms25010356_

Round 1
Reviewer 1 Report
Comments and Suggestions for Authors
The work by L. Hellman et al is an extensive study about the cleavage site specificity of channel catfish granzyme-like II expressed by NK cells. The study has been well performed and will be interesting for the scientific community.
Figure 1 Panel A should be shown to be more clear or if it cannot be displayed well, the authors might want to move to Supplementary material.
The Discussion section is mainly centered on the evolution of proteases but the section could be highly improved if it is explained how the performed experimental studies performed drive to the evolutionary conclusions.
Author Response
- The figure 1 has a lower resolution in the first draft of the type set manuscript and will have better resolution in the final article. This figure is also essential for the view of the catfish granzyme-like II enzyme as it shows the position of the fish proteases in the evolution of the vertebrate hematopoietic serine proteases where the majority of the fish proteases (except granzymes A/K) form a separate branch and where the three isolated catfish enzymes are representatives of three major sub-branches within this fish branch.
- We have now presented our view of why the analysis of cleavage specificity and thereby target specificity of fish proteases help us in determining the evolution of the hematopoietic serine proteases and their role in vertebrate immunity, added text marked in red in the end of the discussion.
Reviewer 2 Report
Comments and Suggestions for Authors
The manuscript describes a strategy to characterize the cleavage specificity of catfish granzyme-like II using substrate phage display. The work is interesting and carefully performed. However, there are a few points which need to be addressed by the authors.
1) Abstract: is very long. The authors may reduce the abstract, presenting only the main findings without giving too much details on the results.
2) A general description of the role of NK- 100 like cells have in fish immunity can be added in the introduction.
3) Line 224 – please add a reference to support the term arginine esterase-like
4) Section 2.1. A brief description of the phylogenetic analysis can be added.
5) Results: it is not necessary to use the full name of amino acid sequence – the three letter code is fine.
6) Figure 7. What is the positive and negative control used by the authors for the analysis of the cleavage specificity of zebrafish AE-like using the 2xTrx system ?
7) Minor comments
Line 325 – all bacterial strains should be in italics
Line 362 - in vivo in italics
The volume units is mL or μL not ml or μl.
Author Response
- We have now deleted a part of the abstract where details of the specificity is described to shorten the abstract.
- The exact role of these NK-like cells is not known but seems to act similarly to mammalian NK-cells. Very little work has been done of them except for the nice study by Shen et al 2002, and we refer to this the best study of my knowledge.
- Arginine-esterase is just a name added by the bioinformaticians that set the layout of the genome as they run homology search and the gene they identify the closest homologue gives the name in the annotation even if the protease functionally most often has no similarity. These names therefore often change when more information is becoming available. As can be seen from figure 1 the fish proteases form a separate subfamily in the large tree and mammalian homologues thereby giving little help in defining the function and name of the particular fish protease based on mammalian similarity.
- A brief description of the phylogenetic analysis has been added to section 2.1.
- All amino acids have now been given in three letter code.
- The system acts as internal positive and negative controls. When we take a sequence selected as preferred from the phage display the cleavage is very efficient and when we modify the sequence to separate from the consensus we often see no cleavage. These two examples act as positive and negative controls, as can be seen from figure 4C. Having only one Arg no cleavage and having four Arg we see almost complete cleavage already after 15 minutes.
- All bacterial strains are now in italics.
- In vivo in italics.
- Volumes are now in mL and uL.